# Association between ambient temperature and hypertensive disorders in pregnancy in China

Tao Xiong [1,2,3], Peiran Chen [2,4], Yi Mu [2,4], Xiaohong Li[2,4], Baofeng Di[5,6], Jierui Li[5,6], Yi Qu[1,2], Jun Tang[1,2], Juan Liang [2,4,7✉] & Dezhi Mu [1,2,7✉]

Hypertensive disorders in pregnancy (HDPs) are leading perinatal diseases. Using a national cohort of 2,043,182 pregnant women in China, we evaluated the association between ambient temperatures and HDP subgroups, including preeclampsia or eclampsia, gestational hypertension, and superimposed preeclampsia. Under extreme temperatures, very cold exposure during preconception (12 weeks) increases odds of preeclampsia or eclampsia and gestational hypertension. Compared to preconception, in the first half of pregnancy, the impact of temperature on preeclampsia or eclampsia and gestational hypertension is opposite. Cold exposure decreases the odds, whereas hot exposure increases the odds. Under average temperatures, a temperature increase during preconception decreases the risk of preeclampsia or eclampsia and gestational hypertension. However, in the first half of pregnancy, temperature is positively associated with a higher risk. No significant association is observed between temperature and superimposed preeclampsia. Here we report a close relationship exists between ambient temperature and preeclampsia or eclampsia and gestational hypertension.

[1] Department of Pediatrics, West China Second University Hospital, Sichuan University, 610041 Chengdu, China. [2] Key Laboratory of Birth Defects and Related Diseases of Women and Children (Sichuan University) Ministry of Education, Sichuan University, 610041 Chengdu, China. [3] Deep Underground Space Medical Center, West China Hospital, Sichuan University, 610041 Chengdu, China. [4] National Office for Maternal and Child Health Surveillance of China, Department of Obstetrics, West China Second University Hospital, Sichuan University, Chengdu 610041 Sichuan, China. [5] Institute for Disaster Management and Reconstruction, Sichuan University-The Hongkong Polytechnic University, 610200 Chengdu, Sichuan, China. [6] Department of Environmental Science and Engineering, Sichuan University, 610065 Chengdu, Sichuan, China. [7] These authors contributed equally: Juan Liang, Dezhi Mu. ✉email: liangjuan002@163.com; mudz@scu.edu.cn

Hypertensive disorders in pregnancy (HDPs) are the most common pregnancy complications. Such disorders occur in an estimated 3–8% of pregnancies worldwide, and the incidence has increased over time[1,2]. HDPs place an enormous burden on pregnant women and their offspring and are among the leading causes of maternal and offspring mortality and morbidity, especially in low-income and middle-income settings[3,4]. Among pregnant women, HDPs are strongly associated with pregnancy-related diseases[5] and future cardiovascular, renal and cerebral diseases[6–8]. Regarding the fetus, HDPs are major contributors to premature delivery[9] and stillbirths as demonstrated in our previous studies[10]. Therefore, reducing mortality and morbidity from HDPs is a global priority for women and infant health. Currently, preventive and therapeutic strategies for HDPs are lacking as the mechanism is not completely understood. The risk factors have been extensively studied, and the known factors include inherited susceptibility[11], placental angiogenic dysfunction[12], etc. There is a need for a deep understanding of the pathogenesis of HDPs.

In the context of global climate change, accumulating epidemiological evidence indicates that abnormal ambient temperatures could increase the risk of a wide range of cardiorespiratory diseases[13] and perinatal diseases[14–17]. As HDPs are considered special cardiovascular diseases occurring during the perinatal period, it is possible that the ambient temperature may have an important role in modifying the risk of HDPs. Limited studies detecting the associations between meteorological variables and HDPs have been performed. Most previous studies assessed the effects of seasonal variation on the prevalence of HDPs and obtained remarkably different results. Compared with other seasons, lower prevalence rates of HDPs were reported in women who delivered in the autumn[18,19] or summer[20] or conceived in the autumn[21]. These results suggest that a seasonal driver of HDPs exists that is independent of other factors. The role of the ambient temperature, which is among the most important variables contributing to seasonal variation, in the development of HDPs has been poorly explored. Recently, a pilot study reported an association between the ambient temperature and preeclampsia (a subtype of HDP)[22]. However, the validity of these results is questionable because of potential bias in the analyses[23]. Our primary aim is to investigate the associations between HDPs and ambient temperatures. The secondary aim is to identify possible vulnerable populations with the goal of reducing HDPs and improving maternal and infant perinatal outcomes.

## Results

**Summary statistics**. The sociodemographic characteristics of the included women are shown in Table 1. In total, 2,043,182 pregnant women were included during the study period. The median age is 28 years old (interquartile range 25–31 years old). In total, 1,973,919 pregnant women without complications (96.61%) served as controls. Among the 69,263 women with HDPs (3.39%), 23,704 women had gestational hypertension (1.16%), 38,166 women had preeclampsia or eclampsia (1.87%), 5453 women had chronic hypertension (0.27%), and 1940 women had superimposed preeclampsia (0.10%). Most women with HDPs were from level 2 and level 3 hospitals (level 2 and 3 represent the largest hospitals), had more than 4 antenatal care visits, were married, were in the 25–34 or 35–39-years-old age groups, were nulliparous or with 1 parity, birthed singleton infants, did not birth small for gestational age (SGA) infants, and did not birth preterm infants. The women with HDPs tended to undergo a cesarean section, whereas the normotensive women tended to undergo vaginal delivery. The distributions of the region, fetus's sex, and season of conception were similar in each group.

**Risk associated with extreme temperatures**. Compared with women in the moderate local temperature range, the adjusted associations between extreme ambient temperatures and HDPs in China are presented in Fig. 1. During preconception, very cold exposure (below the 5th percentile) increased the odds of preeclampsia or eclampsia (adjusted odds ratio (aOR): 1.22, 95% confidence interval (CI): 1.12–1.32) (Fig. 1a). In the first half of pregnancy, the impact of extreme temperatures on preeclampsia or eclampsia appeared to be opposite of that during preconception. Very cold (aOR: 0.89, 95% CI: 0.84–0.94) and moderate cold (between the 5th and 10th percentile) (aOR: 0.86, 95% CI: 0.81–0.92) temperature exposures reduced the odds of preeclampsia or eclampsia, whereas very hot (above the 95th percentile) (aOR: 1.16, 95% CI: 1.10–1.22) and moderate hot (between the 90th and 95th percentile) (aOR: 1.13, 95% CI: 1.07–1.19) temperature exposures increased the odds of preeclampsia or eclampsia. Similar associations were observed between extreme ambient temperatures and gestational hypertension (Fig. 1b). No significant association was observed between extreme ambient temperatures and superimposed preeclampsia (Fig. 1c).

When stratified by study site, most results in North and South China were consistent with the trends of the whole country (Table 2), although wider CIs were observed in North and South China due to the decreased sample size. Regarding preeclampsia or eclampsia and gestational hypertension during preconception, cold exposure showed increased odds, whereas hot temperature exposure decreased the odds. In the first half of pregnancy, the impact of extreme temperatures seems to be opposite to that observed during preconception. Regarding superimposed preeclampsia, no significant association was found during most exposure periods in all regions.

**Trends of risk associated with the average temperature**. Based on the average temperature analysis, Fig. 2 illustrates the pooled cumulative exposure–response curves of the associations between the weekly average temperatures and preeclampsia or eclampsia. During preconception (Fig. 2a), decreasing odds were observed with increasing nonoptimum temperatures. However, in the first half of pregnancy, the tendency was generally reversed. During 1–20 weeks (Fig. 2b), 5–12 weeks (Fig. 2d), and 13–20 weeks (Fig. 2e), increasing temperatures were associated with increasing odds of preeclampsia or eclampsia, except for a relative flat curve during 1–4 weeks (Fig. 2c). The curves of gestational hypertension and the average temperature (Fig. 3) were generally similar to those of preeclampsia or eclampsia. Regarding superimposed preeclampsia (Fig. 4), the large CIs of the curves covered the invalid value, and a nonsignificant relationship was found between temperature and superimposed preeclampsia. When stratified by study site per HDP subgroup, the curves of North and South China were consistent with the trends of the whole country (Supplementary Figs. 1–6).

**Risk change in subgroups exposed to extreme temperatures**. Supplementary Figs. 7–11 depict the change in the risk of preeclampsia or eclampsia in subgroups based on the mother's age, education level, number of fetuses, parity, preterm, SGA, region, and season of conception. During preconception (Supplementary Figs 7a–11a) and in the first half of pregnancy (Supplementary Figs. 7b–11b), women who were aged 20–24 or 25–34 years, were highly educated, had singleton births, had low parity, did not have preterm infants, did not have SGA infants, and lived in urban areas were generally more sensitive to extreme ambient temperatures than the women in the other groups. Similar subgroups sensitive to gestational hypertension were

**Table 1 Sociodemographic characteristics of pregnant women.**

| | Normotensive women<br>n = 1,973,919 | Gestational hypertension<br>n = 23,704 | Preeclampsia or Eclampsia<br>n = 38,166 | Chronic hypertension<br>n = 5453 | Superimposed preeclampsia<br>n = 1940 |
|---|---|---|---|---|---|
| **Region** | | | | | |
| East | 585,971 (29.69%) | 8055 (33.98%) | 10,901 (28.56%) | 1938 (35.54%) | 549 (28.30%) |
| Central | 783,049 (39.67%) | 9784 (41.28%) | 16,421 (43.03%) | 1919 (35.19%) | 857 (44.18%) |
| West | 604,899 (30.64%) | 5865 (24.74%) | 10,844 (28.41%) | 1596 (29.27%) | 534 (27.53%) |
| **Hospital level** | | | | | |
| Unknown | 96,297 (4.88%) | 697 (2.94%) | 1774 (4.65%) | 309 (5.67%) | 60 (3.09%) |
| Level 1 | 126,596 (6.41%) | 1116 (4.71%) | 1559 (4.08%) | 175 (3.21%) | 36 (1.86%) |
| Level 2 | 883,819 (44.77%) | 9717 (40.99%) | 12,874 (33.73%) | 2041 (37.43%) | 436 (22.47%) |
| Level 3 | 867,207 (43.93%) | 12,174 (51.36%) | 21,959 (57.54%) | 2928 (53.70%) | 1408 (72.58%) |
| **Antenatal care visits** | | | | | |
| None | 22,224 (1.13%) | 276 (1.16%) | 832 (2.18%) | 74 (1.36%) | 56 (2.89%) |
| 1–3 | 113,037 (5.73%) | 1502 (6.34%) | 2889 (7.57%) | 332 (6.09%) | 175 (9.02%) |
| 4–6 | 581,397 (29.45%) | 6171 (26.03%) | 11,946 (31.30%) | 1319 (24.19%) | 583 (30.05%) |
| 7–9 | 591,251 (29.95%) | 6966 (29.39%) | 11,282 (29.56%) | 1474 (27.03%) | 541 (27.89%) |
| ≥10 | 615,609 (31.19%) | 8078 (34.08%) | 9640 (25.26%) | 2066 (37.89%) | 445 (22.94%) |
| Unknown | 50,401 (2.55%) | 711 (3.00%) | 1577 (4.13%) | 188 (3.45%) | 140 (7.22%) |
| **Mother's education** | | | | | |
| College of higher | 760,655 (38.54%) | 9240 (38.98%) | 13,657 (35.78%) | 2406 (44.12%) | 671 (34.59%) |
| High school | 554,642 (28.10%) | 6500 (27.42%) | 11,304 (29.62%) | 1390 (25.49%) | 579 (29.85%) |
| Middle school | 563,258 (28.54%) | 6462 (27.26%) | 10,331 (27.07%) | 1289 (23.64%) | 495 (25.52%) |
| Primary school | 45,958 (2.33%) | 824 (3.48%) | 1508 (3.95%) | 211 (3.87%) | 99 (5.10%) |
| None | 8147 (0.41%) | 162 (0.68%) | 371 (0.97%) | 34 (0.62%) | 20 (1.03%) |
| Unknown | 41,259 (2.09%) | 516 (2.18%) | 995 (2.61%) | 123 (2.26%) | 76 (3.92%) |
| **Marital status** | | | | | |
| Single/widower/divorced/cohabitation | 24,424 (1.24%) | 281 (1.19%) | 449 (1.18%) | 47 (0.86%) | 15 (0.77%) |
| Married | 1,949,046 (98.74%) | 23,419 (98.80%) | 37,712 (98.81%) | 5405 (99.12%) | 1925 (99.23%) |
| Unknown | 449 (0.02%) | 4 (0.02%) | 5 (0.01%) | 1 (0.02%) | |
| **Mother's age** | | | | | |
| <20 | 38,179 (1.93%) | 359 (1.51%) | 610 (1.60%) | 55 (1.01%) | 12 (0.62%) |
| 20–24 | 313,318 (15.87%) | 2938 (12.39%) | 5118 (13.41%) | 446 (8.18%) | 125 (6.44%) |
| 25–34 | 1,338,331 (67.80%) | 15,064 (63.55%) | 23,648 (61.96%) | 3331 (61.09%) | 1033 (53.25%) |
| 35–39 | 170,312 (8.63%) | 3311 (13.97%) | 5546 (14.53%) | 1015 (18.61%) | 468 (24.12%) |
| ≥40 | 36,739 (1.86%) | 1222 (5.16%) | 1939 (5.08%) | 431 (7.90%) | 246 (12.68%) |
| Unknown | 77,040 (3.90%) | 810 (3.42%) | 1305 (3.42%) | 175 (3.21%) | 56 (2.89%) |
| **Delivery method** | | | | | |
| Vaginal | 1,186,080 (60.09%) | 10,308 (43.49%) | 8365 (21.92%) | 1865 (34.20%) | 292 (15.05%) |
| Cesarean section | 787,839 (39.91%) | 13,396 (56.51%) | 29,801 (78.08%) | 3588 (65.80%) | 1648 (84.95%) |
| **Fetus's gender** | | | | | |
| Female | 949,029 (48.08%) | 11,564 (48.79%) | 19,112 (50.08%) | 2648 (48.56%) | 946 (48.76%) |
| Male | 1,021,145 (51.73%) | 12,108 (51.08%) | 18,827 (49.33%) | 2787 (51.11%) | 964 (49.69%) |
| Unknown | 3745 (0.19%) | 32 (0.13%) | 227 (0.59%) | 18 (0.33%) | 30 (1.55%) |
| **Parity** | | | | | |
| Nulliparous | 1,107,845 (56.12%) | 13,727 (57.91%) | 22,729 (59.55%) | 2868 (52.59%) | 815 (42.01%) |
| 1 | 745,118 (37.75%) | 8232 (34.73%) | 12,539 (32.85%) | 2141 (39.26%) | 895 (46.13%) |
| 2 | 101,917 (5.16%) | 1429 (6.03%) | 2338 (6.13%) | 363 (6.66%) | 166 (8.56%) |
| ≥3 | 18,680 (0.95%) | 311 (1.31%) | 546 (1.43%) | 80 (1.47%) | 62 (3.20%) |
| Unknown | 359 (0.02%) | 5 (0.02%) | 14 (0.04%) | 1 (0.02%) | 2 (0.10%) |
| **Number of fetus** | | | | | |
| Single birth | 1,945,153 (98.54%) | 22,902 (96.62%) | 35,514 (93.05%) | 5253 (96.33%) | 1879 (96.86%) |
| Polyembryony | 28,766 (1.46%) | 802 (3.38%) | 2652 (6.95%) | 200 (3.67%) | 61 (3.14%) |
| **Small for gestational age** | | | | | |
| No | 1,810,371 (91.71%) | 20,657 (87.15%) | 26,990 (70.72%) | 4610 (84.54%) | 1305 (67.27%) |
| Yes | 156,765 (7.94%) | 2975 (12.55%) | 10,777 (28.24%) | 815 (14.95%) | 585 (30.15%) |
| Unknown | 6783 (0.34%) | 72 (0.30%) | 399 (1.05%) | 28 (0.51%) | 50 (2.58%) |
| **Preterm** | | | | | |
| No | 1,844,447 (93.44%) | 21,523 (90.80%) | 25,069 (65.68%) | 4489 (82.32%) | 899 (46.34%) |
| Yes | 129,472 (6.56%) | 2181 (9.20%) | 13,097 (34.32%) | 964 (17.68%) | 1041 (53.66%) |
| **Season of conception** | | | | | |
| Warm (Apr–Sep) | 832,708 (42.19%) | 10,823 (45.66%) | 17,094 (44.79%) | 2494 (45.74%) | 871 (44.90%) |
| Cold (Oct–Mar) | 1,141,211 (57.81%) | 12,881 (54.34%) | 21,072 (55.21%) | 2959 (54.26%) | 1069 (55.10%) |

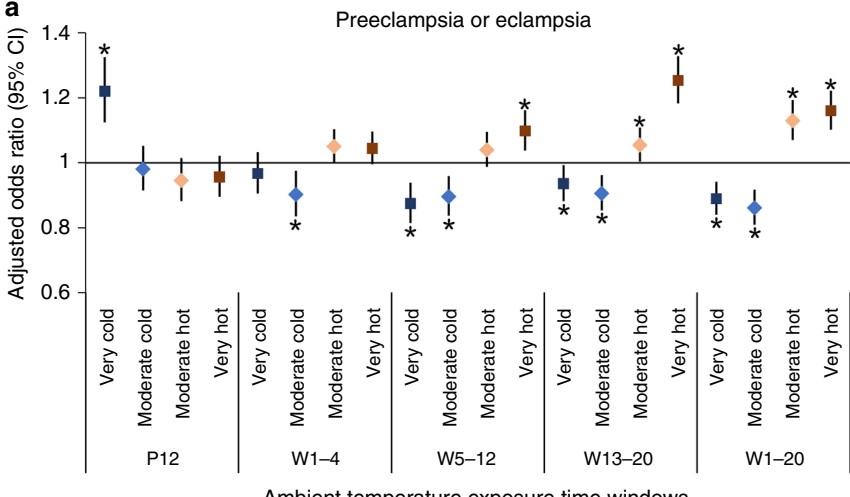

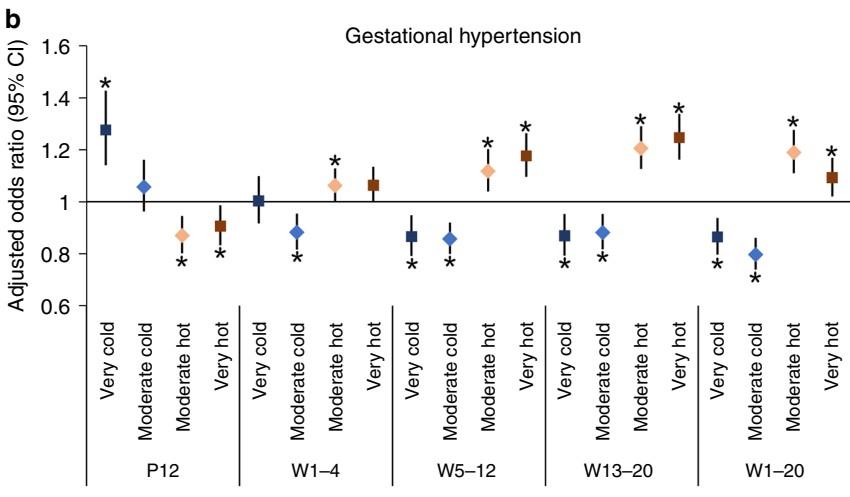

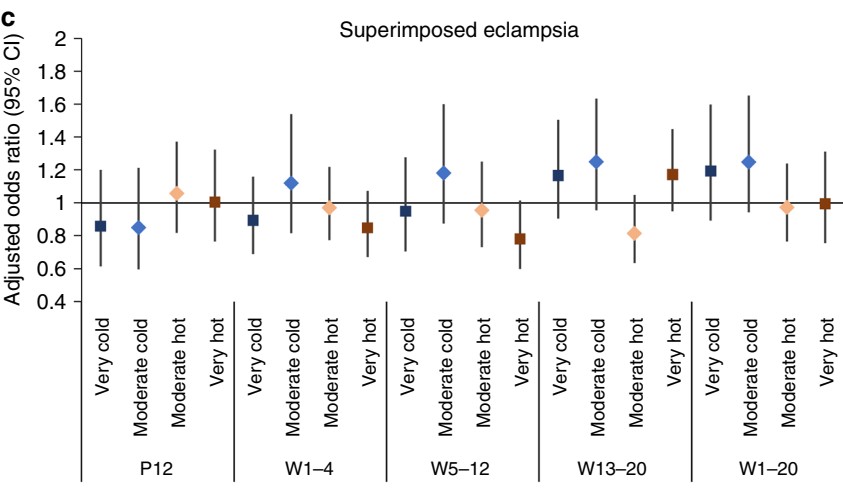

**Fig. 1 Adjusted odds ratios of HDP for extreme temperatures by pregnancy periods.** The models were adjusted for the hospital level, antenatal care visits, marital status, region, mother's education, mother 's age, parity, number of fetuses, elevation, humidity, and air pollution exposure. **a** Preeclampsia or eclampsia; **b** gestational hypertension; **c** superimposed preeclampsia. Data are presented as aORs with 95% confidence intervals. Black error bars correspond to 95% confidence intervals, center for the error bars correspond to points estimate of aORs. Black stars denote *p* < 0.05. PC preconception, W weeks postconception. Source data are provided as a Source Data file.

**Table 2 Odds ratio of extreme temperature exposed and subgroups of hypertensive disorders in pregnancy.**

| Area | Extreme temperature | Time windows | | | | |
|---|---|---|---|---|---|---|
| | | 12 ws before conception | 1–4 ws after conception | 5–12 ws after conception | 13–20 ws after conception | 1–20 ws after conception |
| **Preeclampsia or eclampsia** | | | | | | |
| North | Very cold | 1.17 [1.07,1.29] | 0.96 [0.88,1.04] | 0.83 [0.75,0.93] | 0.93 [0.86,1.02] | 0.91 [0.84,0.99] |
| | Moderate cold | 1.05 [0.95,1.15] | 0.94 [0.85,1.05] | 0.91 [0.84,1.00] | 0.88 [0.81,0.96] | 0.87 [0.79,0.96] |
| | Moderate hot | 0.97 [0.87,1.08] | 1.05 [0.99,1.11] | 1.00 [0.93,1.08] | 1.00 [0.93,1.08] | 1.11 [1.03,1.20] |
| | Very hot | 0.98 [0.89,1.08] | 1.07 [1.00,1.13] | 1.05 [0.97,1.13] | 1.24 [1.16,1.32] | 1.14 [1.06,1.23] |
| South | Very cold | 1.10 [0.97,1.25] | 0.95 [0.85,1.05] | 0.91 [0.84,1.00] | 0.94 [0.87,1.02] | 0.87 [0.80,0.94] |
| | Moderate cold | 0.96 [0.87,1.06] | 0.90 [0.81,1.00] | 0.87 [0.79,0.96] | 0.92 [0.85,1.00] | 0.85 [0.79,0.92] |
| | Moderate hot | 0.90 [0.83,0.98] | 1.04 [0.96,1.13] | 1.09 [1.02,1.18] | 1.11 [1.03,1.18] | 1.16 [1.08,1.25] |
| | Very hot | 0.92 [0.85,1.00] | 1.03 [0.96,1.10] | 1.16 [1.08,1.25] | 1.22 [1.10,1.35] | 1.18 [1.10,1.27] |
| **Gestational hypertension** | | | | | | |
| North | Very cold | 1.31 [1.14,1.50] | 1.00 [0.88,1.14] | 0.97 [0.86,1.10] | 0.76 [0.66,0.87] | 0.95 [0.85,1.06] |
| | Moderate cold | 0.97 [0.83,1.13] | 0.84 [0.75,0.95] | 0.89 [0.81,0.98] | 0.83 [0.74,0.94] | 0.81 [0.71,0.91] |
| | Moderate hot | 0.94 [0.85,1.05] | 1.09 [0.99,1.19] | 1.12 [1.00,1.25] | 1.09 [0.98,1.21] | 1.15 [1.04,1.27] |
| | Very hot | 0.98 [0.87,1.10] | 1.04 [0.95,1.14] | 1.13 [1.03,1.24] | 1.18 [1.08,1.29] | 1.06 [0.96,1.17] |
| South | Very cold | 1.15 [0.97,1.36] | 0.99 [0.87,1.12] | 0.78 [0.69,0.89] | 0.95 [0.85,1.07] | 0.80 [0.71,0.90] |
| | Moderate cold | 1.13 [1.00,1.28] | 0.93 [0.84,1.04] | 0.84 [0.76,0.93] | 0.91 [0.83,1.01] | 0.79 [0.71,0.87] |
| | Moderate hot | 0.81 [0.72,0.91] | 1.05 [0.97,1.14] | 1.13 [1.03,1.24] | 1.30 [1.19,1.42] | 1.23 [1.12,1.35] |
| | Very hot | 0.85 [0.75,0.96] | 1.11 [1.01,1.21] | 1.23 [1.11,1.37] | 1.29 [1.16,1.44] | 1.12 [1.03,1.23] |
| **Superimposed eclampsia[a]** | | | | | | |
| North | Very cold | 0.68 [0.42,1.10] | 0.91 [0.65,1.28] | 0.76 [0.47,1.21] | 1.25 [0.87,1.80] | 1.07 [0.68,1.67] |
| | Moderate cold | 0.80 [0.50,1.28] | 0.88 [0.56,1.38] | 0.99 [0.65,1.52] | 1.06 [0.70,1.59] | 1.21 [0.82,1.77] |
| | Moderate hot | 1.03 [0.75,1.43] | 0.96 [0.70,1.32] | 1.21 [0.84,1.76] | 1.08 [0.77,1.50] | 1.05 [0.75,1.46] |
| | Very hot | 0.94 [0.63,1.41] | 0.72 [0.52,0.99] | 0.71 [0.50,1.00] | 1.49 [1.07,2.06] | 1.11 [0.72,1.70] |
| South | Very cold | 0.92 [0.56,1.51] | 0.88 [0.57,1.38] | 1.12 [0.77,1.62] | 1.22 [0.88,1.69] | 1.39 [0.92,2.08] |
| | Moderate cold | 0.90 [0.52,1.54] | 1.40 [0.86,2.26] | 1.41 [0.92,2.15] | 1.38 [0.96,1.97] | 1.20 [0.81,1.79] |
| | Moderate hot | 1.04 [0.68,1.59] | 1.02 [0.72,1.44] | 0.73 [0.47,1.13] | 0.61 [0.41,0.91] | 0.92 [0.64,1.32] |
| | Very hot | 1.10 [0.76,1.58] | 1.07 [0.73,1.55] | 0.86 [0.59,1.24] | 0.97 [0.70,1.33] | 0.88 [0.60,1.30] |

[a]Take chronic hypertension as reference.

observed when exposed to extreme temperatures (Supplementary Figs. 12–16).

## Discussion

In the context of global climate change, we estimated the contributions of extreme/average temperatures to the risk of HDPs using Chinese national cohort data. Under extreme temperatures, exposure to very cold temperatures before conception and very/moderate hot temperatures in the first half of pregnancy appeared to be associated with increased odds of developing preeclampsia or eclampsia and gestational hypertension, whereas exposure to very/moderate cold temperatures in the first half of pregnancy seemed to decrease the odds of preeclampsia or eclampsia and gestational hypertension. Under average temperatures, an increase in the temperature before conception was associated with a decreased risk of preeclampsia or eclampsia and gestational hypertension. However, in the first half of pregnancy, the temperature was positively associated with the risk of preeclampsia or eclampsia and gestational hypertension; a high temperature increased the odds of preeclampsia or eclampsia and gestational hypertension. Notably, extreme temperatures and the average temperature generally did not have an obvious impact on superimposed preeclampsia. Possible modifiers may affect the association between temperature and HDPs; women who were aged 20–24 or 25–34 years, were highly educated, had singleton births, had low parity, did not have preterm infants, did not have SGA infants, and lived in urban areas were generally sensitive to the ambient temperature. In summary, these results suggest that the ambient temperature may have long-term and chronic effects on preeclampsia or eclampsia and gestational hypertension.

To the best of our knowledge, this study is the first to investigate the effects of the ambient temperature on the risk of HDPs, including preeclampsia or eclampsia, gestational hypertension, and superimposed preeclampsia. To assess HDP development, we developed a county-based dataset covering most areas that represented 8–10% of all pregnant women in China. Multiple phases during pregnancy, including preconception and in the first half of pregnancy, were considered at a high temporal resolution (by week) for the ambient temperature data. The present study provides ample evidence of the relationship between the ambient temperature and risk of HDPs among subgroups. One strength of this study is the inclusion of a pregnant population without other recorded diseases (only HDPs vs. normal pregnancy). The integrity of the included population increased the accuracy of the results related to the actual pathogenesis. The large sample from the national cohort in this study provided an opportunity to evaluate the subtle effects of temperature. The following two possible premises were proposed based on different hypotheses: (1) the dramatic change in temperature induced the risk of HDPs; thus, we determined the effect of extreme temperatures (cold and hot); and (2) temperature modified the risk of HDPs; thus, we calculated the impact of the average temperature on HDPs. We carefully considered possible biases, such as humidity and air pollution exposure. Considering that air conditioning or central

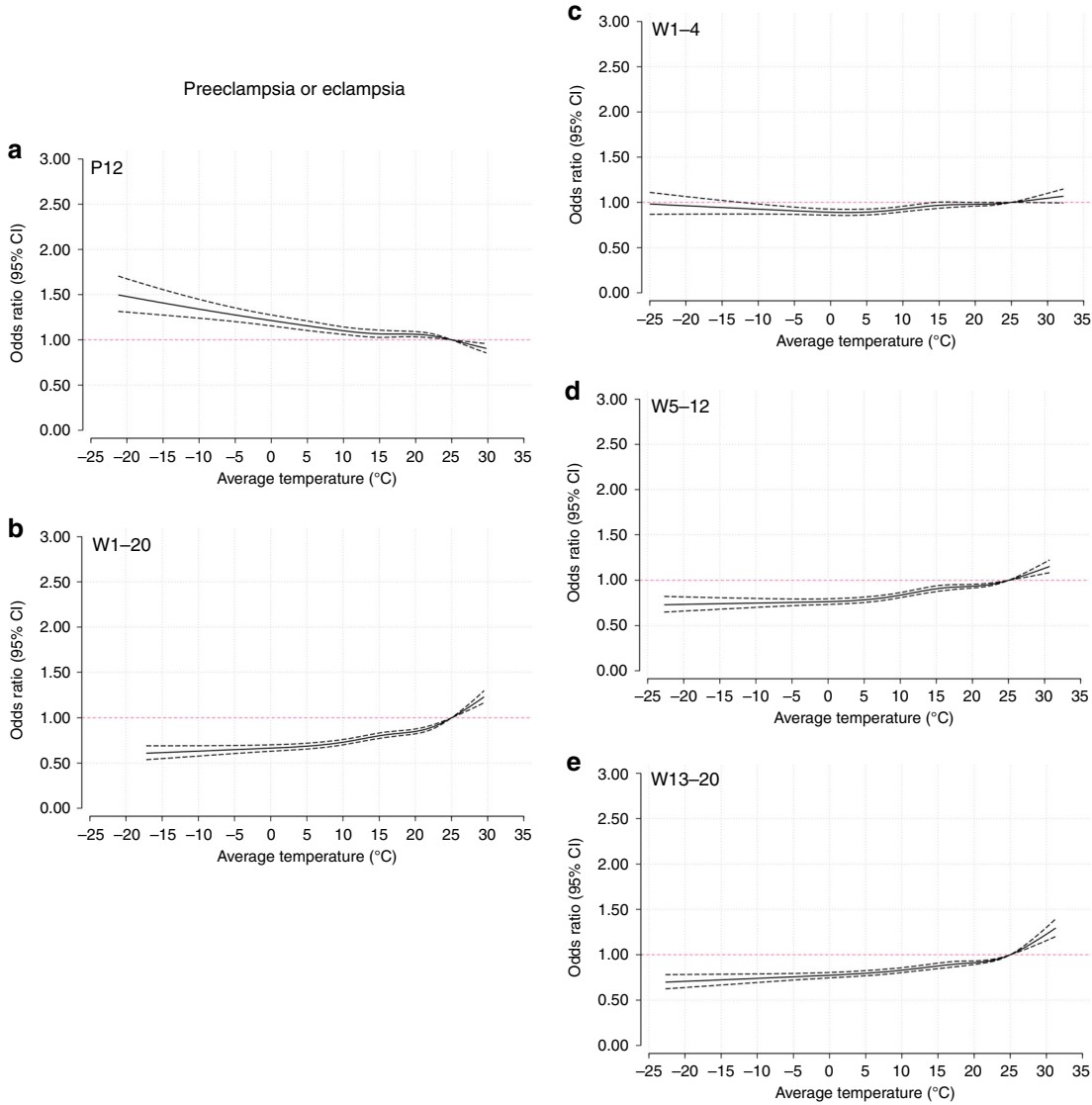

**Fig. 2 Curves of the associations between average temperatures and preeclampsia or eclampsia.** Solid black lines correspond to points estimate of aORs of preeclampsia or eclampsia; dashed black lines correspond to 95% confidence intervals. **a** 12 weeks preconception; **b** 1–20 weeks postconception; **c** 1–4 weeks postconception; **d** 5–12 weeks postconception; **e** 13–20 weeks postconception. P preconception, W weeks postconception. Source data are provided as a Source Data file.

heating in China (as a developing country) is not as common as in developed countries, the Chinese cohort population may be easily influenced by ambient temperature changes and may be more representative for ambient temperature research than those from developed countries. In particular, we performed subgroup analyses by dividing the cohort into North China and South China groups based on the 0 °C isotherm in January, which is the cutoff line for the central heating system. The results from North China and South China generally coincided, demonstrating the weak influence of the heating system on HDPs. Furthermore, temperature exposure over a long period, i.e., from 12 weeks preconception to the first half of pregnancy, was examined. The long period provided adequate time to observe the cumulative effects of chronic exposure to ambient temperatures. Finally, we identified possible modifiers of the association between temperature and HDPs. Women who were aged 20–24 or 25–34 years, were highly educated, had singleton births, had low parity, did not birth preterm infants, did not birth SGA infants, and lived in urban areas were generally sensitive to ambient temperatures. These possible sensitive subpopulations

should be given more attention regarding the influence of extreme temperature exposure. In general, the reported effects of temperatures on HDPs in our study may be pronounced and solid.

Seasonality (i.e., warm and cold seasons) was not included as an adjusted factor in our two analyses (average temperature and extreme temperature) because collinearity exists between season and ambient temperature after we controlled for seasonality in this study. For example, the warm season was closely related to hot temperatures, and the cold season was closely related to cold temperatures. Therefore, the effects of temperature on HDPs were eliminated after controlling for seasonality. To the best of our knowledge, seasons are usually set as subgroups in acute exposure to temperature[24] rather than chronic exposure[25], which is consistent with this study. Therefore, we added more analyses and used two seasons as a subgroup of the acute exposure to temperature (Supplementary Figs. 11 and 16). Interestingly, we found that the effects of temperature during the warm and cold seasons were similar to those in the whole population, suggesting that the outcome is robust in different seasons.

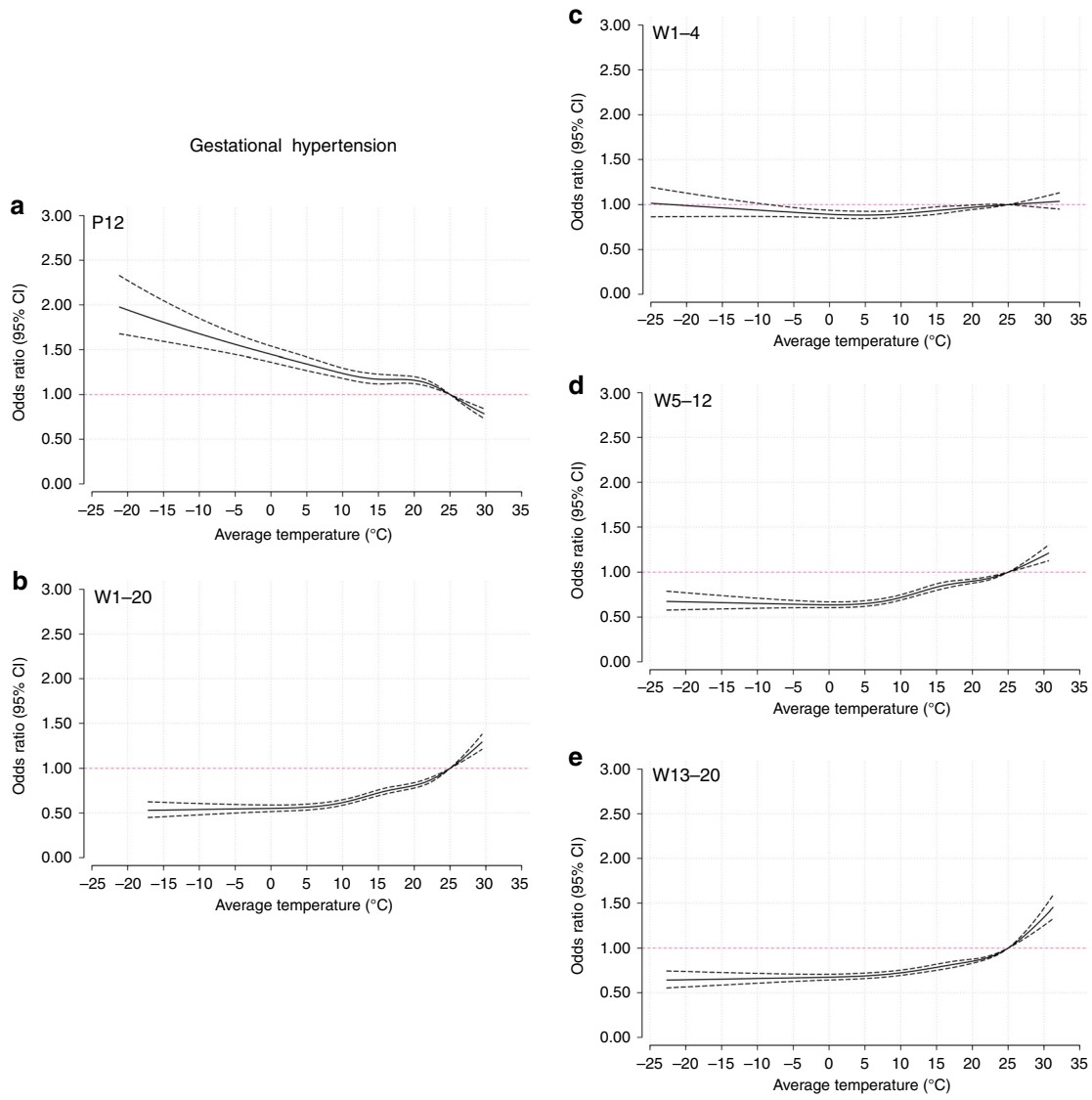

**Fig. 3 Curves of the associations between average temperatures and gestational hypertension.** Solid black lines correspond to points estimate of aORs of gestational hypertension; dashed black lines correspond to 95% confidence intervals. **a** 12 weeks preconception; **b** 1–20 weeks postconception; **c** 1–4 weeks postconception; **d** 5–12 weeks postconception; **e** 13–20 weeks postconception. PC preconception, W weeks postconception. Source data are provided as a Source Data file.

To date, only one pilot study explored the effects of ambient temperatures on the risk of preeclampsia in Canada[22]. The risk of preeclampsia among those who experienced hot temperatures at conception and cold temperatures at the end of pregnancy was increased. However, the association between preeclampsia and temperature was invalidated by the adjustment of the length of gestation. The authors speculated that the associations between ambient temperatures and preeclampsia may be biased by short gestation periods. Several limitations existed in that study. First, the study captured the associations with temperature during only a 4-week exposure period after conception and before admission. Because the temperature exposure period critical for increasing the risks of HDPs is unclear, multiple lag times and long temperature exposure periods may be necessary[23]. Second, it has been reported that prenatal exposure to air pollution increases gestational hypertension and preeclampsia risks[25,26]. Air pollutants, representing potential confounders, were not adjusted in that study. In our study, to avoid bias due to the length of gestation, we selected equal and multiple lag times of the exposure period for each individual (from 12 weeks preconception to the first half

of pregnancy). These periods provide enough time to detect the subtle and long-term effects of ambient temperatures and avoid bias due to the length of gestation. Furthermore, air pollution exposure was splined as a covariate in our study.

The mechanism by which ambient temperatures influence HDPs remains poorly understood. Cold exposure results in peripheral vasoconstriction and elevated heart rate and blood pressure by activating both the sympathetic nervous system and renin–angiotensin system[27,28]. In addition, cold exposure increases cardiovascular risk biomarkers, including inflammation, coagulation, oxidative stress, endothelial function[29], and cholesterol levels[30]. These changes may be associated with excess risk of cardiovascular disease and contribute to the development of HDPs. Hot exposure results in water and electrolyte loss, increased skin blood flow, falling preload, and underlying hypotension[27,28]. Previous research in the general (nonpregnant) population has shown that cold exposure is associated with increased blood pressure and an increased prevalence of hypertension[31–33]. These results are consistent with our results regarding the risk of HDPs during the preconception period.

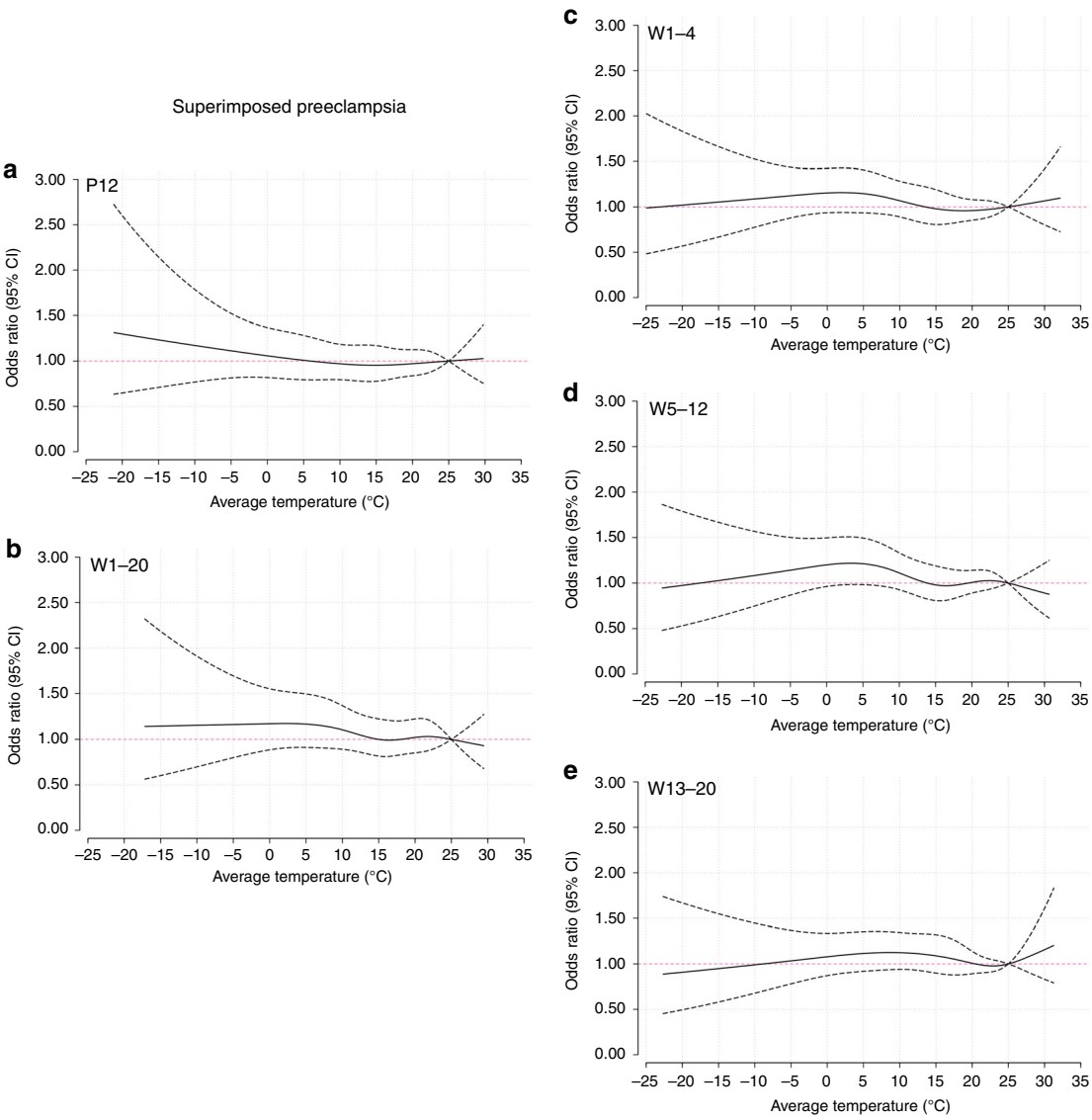

**Fig. 4 Curves of the associations between average temperatures and superimposed preeclampsia.** Solid black lines correspond to points estimate of aORs of superimposed preeclampsia; dashed black lines correspond to 95% confidence intervals. **a** 12 weeks preconception; **b** 1–20 weeks postconception; **c** 1–4 weeks postconception; **d** 5–12 weeks postconception; **e** 13–20 weeks postconception. PC preconception, W weeks postconception. Source data are provided as a Source Data file.

However, in the first half of pregnancy, the effects of temperature on HDPs in this study were reversed a follows: hot exposure is a harmful factor of HDPs, whereas cold exposure has a protective role. A plausible explanation is that gestational changes in thermoregulation render pregnant women vulnerable to hot exposure[34]. First, the weight gained during pregnancy leads to difficulty in heat dissipation. Second, the increased metabolism due to fetus growth results in increased core body temperature in pregnant women. Thus, pregnant women are vulnerable to hot exposure due to the decrease in the capacity of heat loss and the increase in internal heat production[35]. Increased vasoconstrictive reactivity is characteristic of HDPs because the sympathetic nervous system becomes overactive in response to stimuli, including temperature[36]. Hot exposure could disturb thermoregulation in pregnant women, inducing the activation of the sympathetic nervous system and increasing the risk of HDPs. In contrast, cold exposure may help balance thermoregulation in pregnant women in the first half of pregnancy, preventing the activation of the sympathetic nervous system and reducing the risk of HDPs.

Our division of the first half of pregnancy is based on windows critical for HDPs based on biological rationale. The following three time windows in the first half of pregnancy were set: 1–4 weeks, corresponding to embryo implantation; 5–12 weeks, corresponding to vascularization and placentation; and the remaining 8 weeks (13–20 weeks). Our study found that the influence of temperature on preeclampsia or eclampsia and gestational hypertension was more obvious during 5–12 weeks (vascularization and placentation) and 13–20 weeks (after placentation) than that during 1–4 weeks (embryo implantation) (Figs. 1a–b, 2c–e, and 3c–e). Placental vascular dysfunction is an essential mechanism for the development of HDPs. We speculated that the influence of temperature on preeclampsia or eclampsia and gestational hypertension may be partially attributable to abnormal placentation and the failure of trophoblast invasion into the placental bed, which are the core mechanisms of the pathogenesis of HDPs[37,38].

As a stimulus, the ambient temperature may modulate vaso-constrictive reactivity and contribute to the risk of preeclampsia or eclampsia and gestational hypertension. Interestingly, pre-eclampsia or eclampsia and gestational hypertension showed similar response patterns to temperature, which is similar to the response to other risk factors[39]. This finding may imply that the pathogenesis of preeclampsia or eclampsia and gestational hypertension is similar. However, superimposed preeclampsia did not seem to be associated with temperature. The diverse response of the HDP subgroups to ambient temperature may be explained by the following recognized concept: the pathogenesis of super-imposed preeclampsia differs from that of preeclampsia or eclampsia and gestational hypertension. Superimposed pre-eclampsia tends to be associated with severe cardiovascular abnormalities[40,41] that may respond differently to temperature.

Several limitations should be acknowledged. The first limitation of this study was that the acute effect of ambient temperature on HDPs was not assessed because the specific time of HDP onset was unavailable. The second limitation of this study was the unavailability of information regarding the indoor heating system. The effect of indoor temperatures on the HDP risk should not be ignored, especially under extreme weather conditions (hot or cold). To minimize the impact of the indoor temperature, we calculated the odds of developing HDPs using meteorological data at the county level. This high geographical resolution guaranteed a similar heating system among the individuals. These comparisons minimized the impact of different indoor heating characteristics. Furthermore, to detect the potential effect of heating, we performed subgroup analyses (North and South China) due to the availability of central heating. The results from North China and South China exhibited similar tendencies with some subtle differences, supporting the reliability of the results. Third, some potential confounders, such as preeclampsia in previous pregnancies, body mass index, and seasonal confounders (e.g., vitamin D, hours of sunlight, changes in diet and physical activity, and changes in employment) were not included as adjusted factors. Although these factors have been reported as possible risk factors for HDPs, our NMNMSS database did not cover these variables.

The associations between ambient temperatures and HDP risks were found in this study through a multi-factorial analysis after adjusting for limited sociodemographic covariates, obstetric covariates, and meteorological covariates. Further studies should address whether indoor temperature management can modify the risk of preeclampsia or eclampsia and gestational hypertension. Such a strategy may reduce HDP morbidity and medical resource consumption by facilitating the control of HDPs in vulnerable pregnant women. This study also highlights the need to determine the acute association between ambient temperatures and the HDP risk. Finally, the associations were pronounced among mothers who were aged 20–24 or 25–34 years, were highly educated, had singleton births, had low parity, did not have preterm infants, did not have SGA infants, and lived in urban areas. These subgroups represent vulnerable subpopulations that require extra precautions.

## Methods

**Data and study population.** Individual maternal data were collected from China's National Maternal Near Miss Surveillance System (NMNMSS). The NMNMSS collects the sociodemographic and obstetric information of pregnant and post-partum women from obstetric departments. The collected data included the patients' names, hospital codes, dates of delivery, numbers of antenatal visits, maternal education level, marital status, maternal age, delivery mode, fetus's gender, parity, and numbers of fetuses. The sampling strategy, data collection and quality control procedures have been detailed elsewhere[42,43]. The surveillance system is broadly representative of China and covers 441 member hospitals that manage more than 1000 deliveries annually. The included member hospitals are

located in 326 districts or counties throughout 30 provinces in mainland China, excluding Tibet. Since the establishment of China's NMNMSS in October 2010, the collected data have been widely used for policy development and disease burden assessments in China and worldwide[10,42,44]. Based on the hospitals' location, we defined the region as eastern, central, or western. The hospital level was defined according to a comprehensive standard that includes the numbers of beds and medical staff, clinical department categories, types and quantity of medical equipment, and funding of the hospitals.

**Meteorological demarcation.** The meteorological data were collected at the county level. We derived the weekly temperature and humidity data from the National Meteorological Information Center (http://data.cma.cn/). The air quality index (AQI) was obtained considering 6 pollutants, i.e., CO, $NO_2$, $SO_2$, $O_3$, $PM_{10}$, and $PM_{2.5}$, and the data were downloaded from the present Ministry of Ecology and Environment of the People's Republic of China (MEE); the MEE originated from the Ministry of Environmental Protection (http://datacenter.mee.gov.cn/websjzx/queryIndex.vm).

We categorized temperature into five groups according to the local temperature at the county level[14–16] as follows: 1. very cold (below the 5th percentile); 2. moderate cold (between the 5th and 10th percentile); 3. moderate (between the 10th and 90th percentile); 4. moderate hot (between the 90th and 95th percentile); and 5. very hot (above the 95th percentile). Humidity and air pollution exposure were defined similarly.

The Qin-Huai line, which traces the Huai River and Qin Mountains near latitude 33 °N, is a natural boundary used for regional demarcation between North and South China[45]. As the Qin-Huai line corresponds to the 0 °C isotherm in January, it is widely considered a temperate line and was used as a cutoff for the implementation of central heating systems. In North China, a central heating system was established for residential urban areas, whereas individual heating is used in southern urban areas and all rural areas[46].

**Outcome definition.** We restricted the analysis to pregnant women who gave birth at or after 20 weeks of gestation. Pregnant women without complications were considered the reference group. Pregnant women with HDPs in the NMNMSS were categorized into the following four mutually exclusive subgroups according to the 2013 ACOG guidelines: (i) gestational hypertension, (ii) preeclampsia or eclampsia, (iii) chronic hypertension, and (iv) superimposed preeclampsia[9]. Gestational hypertension was defined as new-onset hypertension (≥140/90 mmHg) after 20 weeks of gestation with the normalization of blood pressure at 12 weeks postpartum. Preeclampsia was defined as hypertension (≥140/90 mmHg) and proteinuria after 20 weeks of gestation or hypertension plus the involvement of one organ or system in women with previously normal blood pressure. Eclampsia was diagnosed as the presence of new-onset grand mal seizures in women with pre-eclampsia. Chronic hypertension was defined as hypertension (≥140/90 mmHg) before pregnancy or before 20 weeks of gestation. Superimposed preeclampsia was defined as chronic hypertension associated with preeclampsia.

We excluded pregnant women with (i) other recorded obstetric complications (uterine rupture, placenta previa, abruption placentae, placental retention, uterine inertia and puerperal infection, abortion-related bleeding and infection) or (ii) medical complications [heart disease, embolism, hepatopathy, severe anemia (hemoglobin concentration <70 g/L), renal disease (including urinary tract infection and chronic kidney disease), lung disease (including upper respiratory tract infection), diabetes (including gestational diabetes mellitus), HIV, desmosis, cancer, etc]. The exclusion criteria were pre-established. In summary, after excluding pregnant women with any of the above diseases, we included pregnant women with only HDPs and healthy pregnant women (control group).

**Time window of ambient exposure.** HDPs occur between the antepartum and postpartum periods. Because the identification and diagnosis of HDPs are usually based on antenatal care visits or severe HDP symptoms, the accurate time of HDP onset was difficult to identify. Only chronic exposure rather than acute exposure to ambient temperatures could be calculated in the women with HDPs. The ambient temperature exposure times of the women with HDPs were classified as pre-conception (12 weeks before conception, because other research concerning temperature exposure and perinatal outcomes[14,15] and prior research investigating air pollution and HDPs[25] generally selected 12 weeks as a preconceptional time window) and in the first half of pregnancy (1–20 weeks of gestation since the diagnosis of gestational hypertension, preeclampsia/eclampsia, and superimposed preeclampsia is established after 20 weeks of gestation). The first half of pregnancy was also subdivided into the following three groups: 1–4 weeks of gestation (embryo implantation), 5–12 weeks of gestation (vascularization and placentation), and 13–20 weeks of gestation (after placentation)[38].

The relationship between chronic hypertension and the ambient temperature was not calculated because the onset of chronic hypertension may not have been during pregnancy, and the exposure period was unclear. Chronic hypertension was considered the control for superimposed preeclampsia according to the definition of superimposed preeclampsia[9].

**Statistical analyses**. The Individual maternal data was based on delivery date of offsprings, which were from Jan 2012 to Dec 2017. The date of conception was calculated by the delivery date of offsprings according to their gestational age. Then, we counted 12 weeks before conception and the first half of pregnancy to calculate the exposure duration (total 32 weeks). The complete meteorological data were available from Oct 2010 to Dec 2016. We matched the individual maternal data and meteorological data using exposure duration. Finally, the date of conception from Dec 2013 to Jul 2016 was considered in this study.

For each individual case, we calculated the average temperature, average humidity and average air pollution exposure during the 12 weeks before conception, 1–4 weeks, 5–12 weeks, 13–20 weeks, and 1–20 weeks in the first half of pregnancy as the exposure periods. Considering the heating supply in the north, we stratified the women into the north and south groups according to the Qin-Huai line. The covariates used in the adjusted model included region, hospital level, number of antenatal visits, maternal education, marital status, maternal age, parity, number of fetuses, elevation, humidity, and air pollution exposure.

We separately analyzed the relationships between temperature and the different subgroups of HDPs according to extreme temperatures and average temperatures. In the extreme temperature analysis, a logistic regression was used to analyze the association between exposure to very cold, moderate cold, moderate hot, and very hot and HDPs considering the cluster effect at the county level. Moderate temperature was used as a reference, and the aOR and 95% CI was used to estimate the strength of the association between the temperatures and HDPs. Categorized humidity, air pollution exposure and other covariates were used to adjust the model.

In the average temperature (weekly) analysis, a random intercept multi-level (county is the high level) logistic regression model was used. We included the restricted cubic spline (RCS) with 5 knots placed at the 5th, 25th, 50th, 75th, and 95th[47] percentiles to allow nonlinear assumptions between the temperature and HDPs, and we plotted the estimated aOR with the 95% CI to show the relationship and its change according to temperature.

Humidity and air pollution exposure were included in the model as covariates after the same treatment as temperature (extreme temperature and average temperature). Other covariates were included as category variables in the model. All analyses were stratified by north and south and were performed for five gestational periods.

Subgroup analyses based on demographic (maternal education, age, region, and season of conception) and obstetric (parity, number of fetus, preterm and SGA) characteristics stratified as described in the extreme temperature analysis models were performed. The covariates included sociodemographic variables and meteorological variables (elevation, humidity, and air pollution exposure). The details of the subgroup analyses were shown in Supplementary Table 1.

All statistical analyses were performed using SAS statistical software version 9.4 (SAS Institute Inc., NC, USA) and Stata version 15.1 (Stata Corp., TX, USA). The figures representing the subgroup analysis were generated using R version 3.6.1 (R Foundation for Statistical Computing, http://www.r-project.org). Findings were considered significant at $P < 0.05$ (two-sided).

**Ethics**. The NMNMSS was established by the National Health Commission of the People's Republic of China to improve the quality of maternal and child health in China. The NMNMSS was approved by the Ethics Committee of West China Second University Hospital, Sichuan University, China (Protocol ID: 2012008), and followed the tenets of the Declaration of Helsinki. Patient consent was collected by the surveillance hospitals of NMNMSS when patient was admitted to hospital.

In addition to the establishment of the NMNMSS, the ethical approval (Protocol ID: 2012008) also permit use of data for following studies (including current study) on maternal health from the NMNMSS.

**Reporting summary**. Further information on research design is available in the Nature Research Reporting Summary linked to this article.

## Data availability

The data that support the findings of this study are available from National Office for Maternal and Child Health Surveillance of China but restrictions apply to the availability of these data, which were used under license for the current study, and so are not publicly available. Data are however available from the authors upon reasonable request and with permission of National Health Commission of the People's Republic of China. The source data underlying Figs. 1a–c, 2a–e, 3a–e and 4a–e and Supplementary Figs. 1a–e, 2a–e, 3a–e, 4a–e, 5a–e, 6a–e, 7a, 8a, b, 9a, b, 10a, b, 11a, b, 12a, b, 13a, b, 14a, b, 15a, b and 16a, b are provided as a Source Data file. Source data are provided with this paper.

## Code availability

The codes used for the statistical models in the main-text are recorded as Supplementary Software 1 (including a Stata code file of RCS analysis for average temperature analysis, a SAS code file for extreme temperature analysis, a R code file for drawing forest plot of subgroup analysis results, and an introduction file of variables in original datasets). All codes are available from the corresponding author upon request. Source data are provided with this paper.

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

## Acknowledgements

This study was supported by the National Science Foundation of China (Nos. 81630038, 81971433, 81971428, 81300525, and 41977245), the National Key Research and Development Program (2017YFA0104200), Deep Underground Space Medical (No. DUGM201809), the National Key R&D Program of China (No. 2019YFC1005100), the Major State Basic Research Development Program (2013CB967404 and 2012BAl04B00), the Ministry of Education of China (IRT0935), the Science and Technology Bureau of Sichuan Province (2020YFS0041, 2020YJ0298, 2020YJ0236, and 2016TD0002), and the clinical discipline program (Neonatology) of the Ministry of Health of China (1311200003303).

## Author contributions

T.X., P.C., and Y.M. are joint first authors. T.X., P.C., Y.M., J.Liang, and D.M. designed the study and analyzed the data. B.D. and J.Li collected and analyzed the meteorological data. T.X., P.C., and Y.M. drafted the paper. X.L., Y.Q., and J.T. contributed to the critical interpretation of the results and development of the report. J.Liang and D.M. proofread the manuscript. D.M. financially supported this study.

## Competing interests

The authors declare no competing interests.
