## [Peer Review File · Nature Communications]

Reviewers' comments:

Reviewer #1 (Remarks to the Author):

Thank you for the opportunity to review this manuscript. This study evaluates the relationship of ambient temperature extremes (<5th and >95th percentile) and a flexible spline for temperature with development of a hypertensive disorders of pregnancy (HDPs, gestational hypertension, preeclampsia or eclampsia, and preeclampsia superimposed on chronic hypertension). The authors evaluated exposure both during preconception (12 weeks prior to gestation) and early/mid-pregnancy (1-20 weeks gestation), controlling for humidity, air pollution and other factors. They found low temperatures during preconception and high temperatures in early/mid-pregnancy to be associated with higher risk of HDPs, while high temperatures during preconception and low temperatures in early/mid-pregnancy were associated with lower risk of HDPs. This is an important area of research, as ambient temperature may have broad population-level impacts and findings may help understand future impacts of climate change. However, I have some questions and concerns below, particularly as relates to timing of exposure to etiologically relevant windows and concerns for time-dependent confounding by season.

1. The authors focus on three exposure windows: 12 months before pregnancy (preconception), gestational weeks 1-10 and gestational weeks 11-20 (as well as a combined 1-20 gestational weeks). However, the biologic rationale behind selection of these windows is not clear. In particular, no etiologic reasoning is provided for how temperature exposure prior to establishment of pregnancy would affect risk of hypertension in pregnancy. Additionally, it's clear that there are important windows in the first 20 weeks' of gestation, but it's not clear why a rough division into weeks 1-10 and 11-20 was employed. If effects are particularly important during implantation, vascularization and placentation, why not target these timepoints specifically?

2. My other major concern is the lack of control for seasonality in the models. The authors report that exposure during the preconception window and in early/mid-pregnancy are inverse of each other, but this is what might be expected if season is not controlled for given the two windows are roughly a half-year apart. As the authors note, there's good literature to suggest that cold temperatures result in increases in cardiac output and higher blood pressure – if cold temperatures have an effect late in pregnancy on placental ischemia and/or maternal blood pressure, this effect could also confound findings in early pregnancy and the preconception period (~a year before delivery). There are also other potential seasonal confounders that are unaccounted for (e.g. vitamin D, hours sunlight, changes in diet and physical activity, changes in employment). It would be informative to see the findings accounting for seasonal changes.

3. Similar to point 1 above, it would be helpful to provide more on the proposed mechanisms through which temperature may impact HDPs. In particular, I'm wondering whether hot and cold temperatures may have different physiologic impacts (e.g. cold temperatures are associated with greater cardiac output – what are the primary mechanisms for hot temperatures?). Also, there's a good deal of debate about how the etiology of gestational hypertension and preeclampsia, and even subtypes of preeclampsia, may differ. How might these factors influence key windows of exposure for each condition?

4. The authors describe exposure during pregnancy throughout the manuscript as being "after pregnancy", which suggests postpartum (e.g. Abstract, page 2, line 36). Suggest revising language. Please note that "postpartum" appears to be used in error on page 15, line 391.

5. In the abstract, please note time ranges for specific windows (e.g. 12 weeks before pregnancy for preconception, and GW 1-20 for pregnancy window).

6. There are several places where the manuscript could be shortened to improve clarity and readability. This includes the background information on impacts of HDPs in the first paragraph of

the introduction (line 49-61), the text summarizing prior literature on seasonality of HDPs in the introduction (lines 74-83), the summary of secondary analyses in the results (lines 144-197) and in general the summary of prior literature in the discussion.

7. Results: Add percentages to the count data in paragraph 1 on page 5.

8. Table 1: This needs some formatting changes to make the table more readable given the large counts in each cell. Consider ensuring all count data is similarly right-justified (so all ones places line up), or consider removing the count data and only reporting percentages (keeping counts in headers).

9. Results, pages 5 and 6: Why are percent changes reported instead of odds ratios throughout? I think this is fine for the main effect estimate within a sentence (as done in line 112), but percent changes are misleading for odds ratios given an odds ratio is on a log-odds scale. For example, an odds ratio of 1.25 is the same magnitude as an odds ratio of 0.80, but one would be written as a 25% change and the other a 20% change. Also, please make sure to label "OR" and "95% CI" when they appear in parentheses.

10. Why did the authors assume a non-linear association of humidity and air pollution with hypertension in pregnancy? Particularly for air pollution, if a true linear and dose-response relationship exists for air pollution and hypertensive disorders of pregnancy, only looking at the <5th and >95th percentiles could lead to an incomplete control of air pollution as confounder.

11. Please revise language in the Discussion (page 12, line 310): "...associations... were confirmed..." with something more observant of potential for confounding and other biases.

12. Methods, page 16, line 412: Consider renaming the two ways of categorizing temperature, as "relative temperature" and "absolute temperature" already have well-defined meanings.

Reviewer #2 (Remarks to the Author):

The present study was performed to examine the association between ambient temperature and the risk of hypertensive disorders in pregnancy using a cohort study from China. The study was based on data from the NMMSS in China between May 2014 and December 2017.

The classification of the cold, moderate and hot temperature makes sense, however, considering the size of the data the authors should consider dividing the moderate group into finer groups to assess the dose-effect association in more detail. This is an interesting study, however, there are some concerns that the authors may wish to consider:

In the statistical analysis section, when the authors say, 1-10 weeks after pregnancy, 11-120 weeks after pregnancy etc. (lines 404-5), I assume they mean after conception and therefore these should be defined as 1-10 weeks' gestation and 11-20 weeks' gestation etc. The confounding analysis should be described in more detail. Was this based on an a priori selection of potential confounders or on available data? Why was mode of delivery adjusted for? What is the rationale that mode of delivery could be associated with the exposure and a risk factor of the outcome? One would think that pre-eclampsia could influence the mode of delivery but increasing the risk of C-section but not the other way around. On the other hand, one may argue that preterm delivery may reduce the risk of pre-eclampsia but the likelihood is that preterm birth could a mediator if temperature influences gestational age. Pre-eclampsia in previous pregnancies and body mass index are a major risk factors for pre-eclampsia and they seem to be ignored. In all cases, the authors should think carefully about the confounding analysis and the rationale of adjusting for these factors. Why didn't the exclusion criteria include diabetes and chronic kidney disease, for

example? What definitions were used for the exclusion criteria. The authors may want to perform more detailed analyses on pre-eclampsia according to term vs preterm pre-eclampsia and whether pre-eclampsia occurred with or without small for gestational age.

Considering the temperature was calculated at the county level, how many counties did the study data cover? What is the average size of the county? What is the variation in temperature within counties? In other words, is it possible that using exposure at county level may have led to exposure misclassification bias? This leads to the statistical analysis and whether it is appropriate to use logistic regression and adjust for region and hospital as covariates rather than assessing clustering by these factors? Have the authors considered using a multi-level analysis which would be more appropriate for this type of data?

The Discussion should acknowledge the limitations of the study design and potential residual confounding. Body mass index is an obvious factor that was not accounted for and such factors should be addressed in the limitations section. Is there evidence that the data sources are valid and reliable? Are the methods used for data linkage appropriate and robust?

RESPONSE TO THE REVIEWERS

Reviewers' comments:

Reviewer #1 (Remarks to the Author):

General comments

Thank you for the opportunity to review this manuscript. This study evaluates the relationship of ambient temperature extremes (<5th and >95th percentile) and a flexible spline for temperature with development of a hypertensive disorders of pregnancy (HDPs, gestational hypertension, preeclampsia or eclampsia, and preeclampsia superimposed on chronic hypertension). The authors evaluated exposure both during preconception (12 weeks prior to gestation) and early/mid-pregnancy (1-20 weeks gestation), controlling for humidity, air pollution and other factors. They found low temperatures during preconception and high temperatures in early/mid-pregnancy to be associated with higher risk of HDPs, while high temperatures during preconception and low temperatures in early/mid-pregnancy were associated with lower risk of HDPs. This is an important area of research, as ambient temperature may have broad population-level impacts and findings may help understand future impacts of climate change. However, I have some questions and concerns below, particularly as relates to timing of exposure to etiologically relevant windows and concerns for time-dependent confounding by season.

Specific comments

Question 1: The authors focus on three exposure windows: 12 weeks before pregnancy (preconception), gestational weeks 1-10 and gestational weeks 11-20 (as well as a combined 1-20 gestational weeks). However, the biologic rationale behind selection of these windows is not clear. In particular, no etiologic reasoning is provided for how temperature exposure prior to establishment of pregnancy would affect risk of hypertension in pregnancy.

Response: *Thank you for your comments. The exact time window of preconceptional*

temperature exposure and hypertension remains unclear because the exact mechanism is unknown. Twelve weeks before pregnancy was considered because of the following two reasons:

- 1. Other research concerning temperature exposure and perinatal outcomes used 12 weeks as a preconceptional time window [1,2].*
- 2. Prior research investigating air pollution and HDPs selected 12 weeks as a preconceptional time window [3].*

We included this information in the “Method” section (line 501-504) and updated the references in the reference list in the revised manuscript. (Please see revised version)

References:

- [1] Ha S, Liu D, Zhu Y, Soo Kim S, Sherman S, Grantz KL, et al. Ambient Temperature and Stillbirth: A Multi-Center Retrospective Cohort Study. Environmental health perspectives. 2017;125:067011.*
- [2] Ha S, Liu D, Zhu Y, Kim SS, Sherman S, Mendola P. Ambient Temperature and Early Delivery of Singleton Pregnancies. Environmental health perspectives. 2017;125:453-59.*
- [3] Nobles CJ, Williams A, Ouidir M, Sherman S, Mendola P. Differential Effect of Ambient Air Pollution Exposure on Risk of Gestational Hypertension and Preeclampsia. Hypertension (Dallas, Tex : 1979) 2019; 74(2): 384-90.*

Question 2: Additionally, it’s clear that there are important windows in the first 20 weeks’ of gestation, but it’s not clear why a rough division into weeks 1-10 and 11-20 was employed. If effects are particularly important during implantation, vascularization and placentation, why not target these timepoints specifically?

Response: *We appreciate the reviewer’s suggestion and reconsidered the division of the windows according to the biological rationale. The postconception period was subdivided into the following three groups: 1-4 weeks of gestation (embryo implantation), 5-12 weeks of gestation (vascularization and placentation), and 13-20 weeks of gestation (after placentation). Please see lines 507-510.*

Question 3: My other major concern is the lack of control for seasonality in the models. The authors report that exposure during the preconception window and in early/mid-pregnancy are inverse of each other, but this is what might be expected if season is not controlled for given the two windows are roughly a half-year apart. As the authors note, there's good literature to suggest that cold temperatures result in increases in cardiac output and higher blood pressure – if cold temperatures have an effect late in pregnancy on placental ischemia and/or maternal blood pressure, this effect could also confound findings in early pregnancy and the preconception period (~a year before delivery). There are also other potential seasonal confounders that are unaccounted for (e.g. vitamin D, hours sunlight, changes in diet and physical activity, changes in employment). It would be informative to see the findings accounting for seasonal changes.

Response: *We appreciate the reviewer's good idea.*

Seasonality (i.e., warm and cold seasons) was not included as an adjusted factor in our analyses (average temperature and extreme temperature) in this study because of the following reason:

We found that collinearity exists between the season and ambient temperature after controlling for seasonality in this study. For example, the warm season is closely related to hot temperatures, and the cold season is closely related to cold temperatures. Therefore, the effects of temperature on HDPs will be eliminated using the control for seasonality. To the best of our knowledge, seasons are usually set as subgroups in acute exposures to temperature [1] rather than chronic exposures [2], which is consistent with this study. Therefore, we added an analysis using seasons as subgroups in the acute exposure of temperature (Supplementary Figs 11 and 16). Interestingly, we found that the effects of temperature in the warm and cold seasons were similar to those in the whole population, suggesting that the outcome is robust in different seasons. Please see lines 291-304.

However, we discussed the limitation of not using a control for seasonality in this

study. Please see lines 403-408.

References

[1] Ha S, Liu D, Zhu Y, Sherman S, Mendola P. Acute Associations Between Outdoor Temperature and Premature Rupture of Membranes. *Epidemiology (Cambridge, Mass)* 2018; 29(2): 175-82.

[2] Nobles CJ, Williams A, Ouidir M, Sherman S, Mendola P. Differential Effect of Ambient Air Pollution Exposure on Risk of Gestational Hypertension and Preeclampsia. *Hypertension (Dallas, Tex : 1979)* 2019; 74(2): 384-90.

Question 4: Similar to point 1 above, it would be helpful to provide more on the proposed mechanisms through which temperature may impact HDPs. In particular, I'm wondering whether hot and cold temperatures may have different physiologic impacts (e.g. cold temperatures are associated with greater cardiac output – what are the primary mechanisms for hot temperatures?). Also, there's a good deal of debate about how the etiology of gestational hypertension and preeclampsia, and even subtypes of preeclampsia, may differ. How might these factors influence key windows of exposure for each condition?

Response: *We appreciate the reviewer's good suggestions and reorganized the manuscript as follows:*

First, we provided the proposed mechanisms by which temperature may impact HDPs. Then, we discussed the different physiological impacts of hot and cold temperatures on HDPs (lines 332-343).

Second, we discussed why an inverse effect exists between preconception and postconception (lines 344-358).

Third, we discussed the influence of temperature during each postconceptional period and the possible reasons based on biological rationales (lines 359-371).

Fourth, we addressed the possible differences in the etiology of preeclampsia or eclampsia, gestational hypertension, and superimposed preeclampsia (lines 376-389).

Question 5: The authors describe exposure during pregnancy throughout the manuscript as being “after pregnancy”, which suggests postpartum (e.g. Abstract, page 2, line 36). Suggest revising language. Please note that “postpartum” appears to be used in error on page 15, line 391.

Response: *We appreciate the reviewer’s suggestion. The terms “after pregnancy” and “postpartum” were replaced with ‘postconception’ or ‘after conception’ in the manuscript (lines 33 and 38 in the abstract, line 119 and line 507).*

Question 6: In the abstract, please note time ranges for specific windows (e.g. 12 weeks before pregnancy for preconception, and GW 1-20 for pregnancy window).

Response: *These revisions were performed (lines 31-32 and line 33-34).*

Question 7: There are several places where the manuscript could be shortened to improve clarity and readability. This includes the background information on impacts of HDPs in the first paragraph of the introduction (line 49-61), the text summarizing prior literature on seasonality of HDPs in the introduction (lines 74-83), the summary of secondary analyses in the results (lines 144-197) and in general the summary of prior literature in the discussion.

Response: *We appreciate the reviewer’s suggestion. We shortened the above paragraph. Please see the background information of the impacts of HDPs on lines 45-57, the text summarizing the prior literature concerning the role of seasonality in HDPs in the introduction on lines 71-82, the summary of the secondary analyses in the results on lines 158-228, and in general, a summary of the prior literature in the discussion on lines 305-312, 372-376.*

Question 8: Results: Add percentages to the count data in paragraph 1 on page 5.

Response: *This change was applied as suggested. Please see lines 96-100 on page 5.*

Question 9: Table 1: This needs some formatting changes to make the table more readable given the large counts in each cell. Consider ensuring all count data is similarly right-justified (so all ones places line up), or consider removing the count data and only reporting percentages (keeping counts in headers).

Response: *Thank you for this suggestion. The change was applied as suggested. All count data are similarly right-justified. Please see Table 1.*

Question 10: Results, pages 5 and 6: Why are percent changes reported instead of odds ratios throughout? I think this is fine for the main effect estimate within a sentence (as done in line 112), but percent changes are misleading for odds ratios given an odds ratio is on a log-odds scale. For example, an odds ratio of 1.25 is the same magnitude as an odds ratio of 0.80, but one would be written as a 25% change and the other a 20% change. Also, please make sure to label “OR” and “95% CI” when they appear in parentheses.

Response: *We appreciate the reviewer’s suggestion. We replaced “percent changes” with “OR and 95% CI”. Additionally, “OR” and “95% CI” are labeled in parentheses. Please see lines 114-126 on pages 5 and 6.*

Question 11: Why did the authors assume a non-linear association of humidity and air pollution with hypertension in pregnancy? Particularly for air pollution, if a true linear and dose-response relationship exists for air pollution and hypertensive disorders of pregnancy, only looking at the <5th and >95th percentiles could lead to an incomplete control of air pollution as confounder.

Response: *Actually, we were unsure whether the relationship between humidity and*

air pollution and HDP was linear or non-linear before the analysis. Therefore, we performed the following two attempts:

- 1. The average exposure meteorological data (such as temperature, humidity and air pollution) were included in the restricted cubic spline (RCS) model to explore the association (linear or non-linear) between exposure and outcome. Since we found that the association between temperature and HDP was generally non-linear, we applied the same treatment to the other meteorological data (such as humidity and air pollution) to maintain consistency in all meteorological data in the same model.*
- 2. We appreciate the reviewer's suggestions and classified temperature, humidity and air pollution into five types, i.e., very high, moderate high, moderate, moderate low, and very low values (according to county-specific <5th, [5th,10th), [10th, 90th), [90th,95th), ≥95th percentiles), to obtain detailed results.*

Question 12: Please revise language in the Discussion (page 12, line 310): “...associations... were confirmed...” with something more observant of potential for confounding and other biases.

Response: *We changed ‘The associations between ambient temperatures and HDP risks were confirmed in this study’ to ‘The associations between ambient temperatures and HDP risks were found in this study through a multi-factorial analysis after adjusting for limited sociodemographic covariates, obstetric covariates, and meteorological covariates.’ Please see lines 409-413 on page 16.*

Question 13: Methods, page 16, line 412: Consider renaming the two ways of categorizing temperature, as “relative temperature” and “absolute temperature” already have well-defined meanings.

Response: *The change was applied as suggested. The term “relative temperature” was replaced with “extreme temperature”, and “absolute temperature” was replaced with “average temperature”. Please see revised manuscript.*

Reviewer #2 (Remarks to the Author):

The present study was performed to examine the association between ambient temperature and the risk of hypertensive disorders in pregnancy using a cohort study from China. The study was based on data from the NMNMSS in China between May 2014 and December 2017.

Question 1: The classification of the cold, moderate and hot temperature makes sense, however, considering the size of the data the authors should consider dividing the moderate group into finer groups to assess the dose-effect association in more detail. This is an interesting study, however, there are some concerns that the authors may wish to consider.

Response: *We appreciate the reviewer's suggestion. We divided the 'moderate group (between the 5th and 95th percentile)' into three groups to assess the dose-effect association.*

Thus, we currently categorized temperature into the following five groups according to the local temperature at the county level:

- 1. very cold (below the 5th percentile);*
- 2. moderate cold (between the 5th and 10th percentile);*
- 3. moderate (between the 10th and 90th percentile);*
- 4. moderate hot (between the 90th and 95th percentile); and*
- 5. very hot (above the 95th percentile).*

Please see lines 455-460. (Please see revised version)

Question 2: In the statistical analysis section, when the authors say, 1-10 weeks after pregnancy, 11-20 weeks after pregnancy etc. (lines 404-5), I assume they mean after conception and therefore these should be defined as 1-10 weeks' gestation and 11-20 weeks' gestation etc.

Response: *This change was applied as suggested. Actually, reviewer 1 raised the*

same question. The term 'after pregnancy' was changed to 'after conception'. Please see lines 519, 523-524. We also divided the time window according to question 2 of Reviewer 1. Please see lines 507-510.

Question 3: The confounding analysis should be described in more detail. Was this based on an a priori selection of potential confounders or on available data? Why was mode of delivery adjusted for? What is the rationale that mode of delivery could be associated with the exposure and a risk factor of the outcome? One would think that pre-eclampsia could influence the mode of delivery but increasing the risk of C-section but not the other way around.

***Response:** The confounding analysis was based on available data in the surveilling system and potential biological mechanisms. In this study, the variable of 'Mode of delivery' was out of the exposure of the time window (12 weeks before conception and 1-20 weeks after conception). We agree with you that it should not be associated with exposure and a risk factor of the outcome. Therefore, 'mode of delivery' was not included as a covariate in the revised manuscript.*

Question 4: On the other hand, one may argue that preterm delivery may reduce the risk of pre-eclampsia but the likelihood is that preterm birth could a mediator if temperature influences gestational age.

***Response:** We appreciate the reviewer's comments. In this study, we considered the impact of gestational weeks on HDPs. To avoid the bias caused by the gestational weeks (preterm delivery), we included an equal gestational age of pregnant women during the exposure time window in this study (from 12 weeks preconception to 20 weeks after pregnancy). Please see lines 326-330.*

Question 5: Pre-eclampsia in previous pregnancies and body mass index are a major risk factors for pre-eclampsia and they seem to be ignored. In all cases, the authors

should think carefully about the confounding analysis and the rationale of adjusting for these factors.

Response: *We agree with the reviewer's opinions that pre-eclampsia in previous pregnancies and body mass index are risk factors for pre-eclampsia. In addition, it could have been helpful to analyze these risk factors in this study. However, we could not find these variables in the database, i.e., China's National Maternal Near Miss Surveillance System (NMNMSS), used in this study. We apologize for our inability to analyze these two variables. We discussed this issue as a limitation of our study in the "discussion" section of the revised manuscript. Please see lines 403-408.*

Question 6: Why didn't the exclusion criteria include diabetes and chronic kidney disease, for example? What definitions were used for the exclusion criteria.

Response: *The exclusion criteria included diabetes and chronic kidney disease. Please see lines 488-494.*

Question 7: The authors may want to perform more detailed analyses on pre-eclampsia according to term vs preterm pre-eclampsia and whether pre-eclampsia occurred with or without small for gestational age.

Response: *This is a very good idea. We appreciate the reviewer's suggestion. We performed more detailed analyses based on term vs preterm pre-eclampsia and whether pre-eclampsia occurred with or without small for gestational age. "Small for gestational age" and "preterm" were set as subgroups. Please see Supplementary Fig. 9 and Supplementary Fig. 14.*

Question 8: Considering the temperature was calculated at the county level, how many counties did the study data cover? What is the average size of the county? What is the variation in temperature within counties? In other words, is it possible that using

exposure at county level may have led to exposure misclassification bias? This leads to the statistical analysis and whether it is appropriate to use logistic regression and adjust for region and hospital as covariates rather than assessing clustering by these factors? Have the authors considered using a multi-level analysis which would be more appropriate for this type of data?

Response: *This study covered 324 counties. The median area of the counties is 2058.88 km² (interquartile range: 1249.94 to 3470.72 km²). The meteorological stations are usually located in the county level in China [1]. We could not obtain the meteorological data within counties.*

The location of the meteorological stations is based on geoclimatic characteristics, which represent a small relative meteorological variation. Generally, each county has an average of 1 meteorological station, indicating that the variation in temperature within counties is relatively small.

References

[1] Feng S, Hu Q, Qian W. Quality control of daily meteorological data in China, 1951–2000: a new dataset[J]. *International Journal of Climatology*, 2004, 24(7): 853-870.

In the restricted cubic spline model, we used a random intercept multi-level model (county is the high level) to adjust the clustering within counties. In the logistic regression model (the extreme exposure meteorological data, such as temperature, humidity and air pollution), we used the “PROC SURVEYLOGISTIC” procedure to adjust the clustering of births within counties because the multi-level model cannot converge in SAS.

Question 9: The Discussion should acknowledge the limitations of the study design and potential residual confounding. Body mass index is an obvious factor that was not accounted for and such factors should be addressed in the limitations section.

Response: *We appreciate your suggestions. We addressed these factors in the limitation section. Please see lines 403-411.*

Question 10: Is there evidence that the data sources are valid and reliable? Are the methods used for data linkage appropriate and robust?

Response: *The data sources, i.e., China's National Maternal Near Miss Surveillance System (NMNMSS), were described in detail in our previous published papers [1-3]. The data sources are valid and reliable. Their establishment and quality control have been described in previous papers. Since the establishment of NMNMSS in October 2010, the collected data have been widely used in policy development and disease burden assessments in China and worldwide. Please also see Line 433-438.*

References

- [1] Zhu J, Liang J, Mu Y, et al. Sociodemographic and obstetric characteristics of stillbirths in China: a census of nearly 4 million health facility births between 2012 and 2014. *The Lancet Global health* 2016; 4(2): e109-18
- [2] Liang J, Li X, Kang C, et al. Maternal mortality ratios in 2852 Chinese counties, 1996-2015, and achievement of Millennium Development Goal 5 in China: a subnational analysis of the Global Burden of Disease Study 2016. *Lancet* 2019; 393(10168): 241-52.
- [3] Liang J, Mu Y, Li X, et al. Relaxation of the one child policy and trends in caesarean section rates and birth outcomes in China between 2012 and 2016: observational study of nearly seven million health facility births. *BMJ* 2018; 360: k817.

REVIEWERS' COMMENTS:

Reviewer #1 (Remarks to the Author):

The authors provided a detailed and complete response to the initial critique. This is a well-conducted study on a critical topic in maternal and child health. I have only two minor additional suggestions:

1. In the results, Figure 1 and Table 2 - it would be helpful to include the percentile ranges for "very cold", "cold", etc. - only needed for the first mention in the results.

2. For describing gestational weeks 1-20, consider something similar to "early pregnancy"/"first half of pregnancy"/etc. instead of "after conception" to be consistent with other literature.

Reviewer #2 (Remarks to the Author):

I would like to thank the authors for addressing my comments. I have no further comments.

RESPONSE TO REVIEWERS

Reviewer #1 (Remarks to the Author):

The authors provided a detailed and complete response to the initial critique. This is a well-conducted study on a critical topic in maternal and child health. I have only two minor additional suggestions:

Question 1: In the results, Figure 1 and Table 2 - it would be helpful to include the percentile ranges for "very cold", "cold", etc. - only needed for the first mention in the results.

Response: We appreciate the reviewer's suggestion and include the percentile ranges for each categorized temperature at the first mention in the "results" section. Please see lines 101-110.

Question 2: For describing gestational weeks 1-20, consider something similar to "early pregnancy"/"first half of pregnancy"/etc. instead of "after conception" to be consistent with other literature.

Response: Thank you very much for your suggestion. "First half of pregnancy" was used instead of "after conception" in the revised manuscript.

Reviewer #2 (Remarks to the Author):

I would like to thank the authors for addressing my comments. I have no further comments.

Response: Thank you very much.